# Influence of oral administration of kukoamine A on blood pressure in a rat hypertension model

**Christine A. Butts** *, Duncan I. Hedderley, Sheridan Martell, Hannah Dinnan, Susanne Middlemiss-Kraak, Barry J. Bunn, Tony K. McGhie, Ross E. Lill

The New Zealand Institute for Plant and Food Research Limited, Palmerston North, New Zealand

* chrissie.butts@plantandfood.co.nz

**Data Availability Statement:** All the data is within the manuscript and Supporting Information files.

**Funding:** The authors received no specific funding for this work.

## Abstract

The benefits of lowering blood pressure (BP) are well established for the prevention of cardiovascular disease. While there are a number of pharmaceuticals available for lowering BP, there is considerable interest in using dietary modifications, lifestyle and behaviour changes as alternative strategies. Kukoamines, caffeic acid derivatives of polyamines present in solanaceous plants, have been reported to reduce BP. We investigated the effect of orally administered synthetic kukoamine A on BP in the Spontaneously Hypertensive Rat (SHR) laboratory animal model of hypertension. Prior to the hypertension study, we determined the safety of the synthetic kukoamine A in a single oral dose (5 or 10 mg kg$^{-1}$ bodyweight) 14-day observational study in mice. No negative effects of the oral administration of kukoamine A were observed. We subsequently investigated the effect of daily oral doses of kukoamine A (0, 5, 10 mg kg$^{-1}$ bodyweight) for 35 days using the SHR rat model of hypertension. The normotensive control Wistar Kyoto (WKY) strain was used to provide a baseline for normal BP in rats. We observed no effect of orally administered synthetic kukoamine A on arterial hypertension in this laboratory animal model of hypertension.

## Introduction

Hypertension is one of the leading preventable risk factors for death and disability globally [1, 2]. The management and reduction of hypertension are important for the prevention of cardiovascular diseases including stroke and ischemic heart disease, as well as renal dysfunction and other non-cardiovascular diseases such as dementia, cancer, oral health diseases and osteoporosis [1–4]. There are various blood pressure (BP)-lowering drugs regularly prescribed including thiazide diuretics, angiotensin-converting enzyme (ACE) inhibitors, calcium antagonists, angiotensin receptor blockers (ARBs) and beta-blockers [5, 6]. Dietary modifications can lower BP, preventing the development of hypertension and lowering the risk of hypertension-related health effects [7]. These dietary approaches have included the DASH (Dietary Approaches to Stop Hypertension) and Mediterranean diets, reducing sodium intake, increasing potassium, magnesium or protein intake, and limiting alcohol consumption [7, 8]. Lifestyle

**Competing interests:** The authors have declared that no competing interests exist.

and behavioural approaches such as weight loss, increased physical activity, yoga, meditation, and slow breathing are suggested as important components of an overall strategy to reduce BP [9]. Individual foods or food ingredients that have been found to lower BP and reduce hypertension are flaxseed; higher nitrate-containing vegetables such as beetroot and leafy greens such as spinach; vegetables such as corn, broccoli, cauliflower, Brussel sprouts, cabbage, carrots, and soybeans/tofu; and fruits such as avocado, cantaloupe, blueberries, raisins/grapes, and apples/pears [10–12].

Potato (*Solanum tuberosum* L.) is the third most consumed crop around the world and is grown in most countries [13]. The potato tuber supplies a wide range of nutrients and diverse bioactive compounds that are known to prevent and combat chronic diseases such as hypertension, cancer, diabetes, and heart disease [14–19]. The bioactive compounds found in potatoes are polyphenols (chlorogenic acid, apigenin, rutin, kaempferol rutinoside), terpenes (lutein and neoxanthin), alkaloids (solanine, tomatine, chaconine) and polyamines (kukoamines). A number of these compounds have demonstrated activity against heart disease and hypertension [20–23]. Chaparro et al. [24] observed relatively high levels of kukoamine in cooked russet and chipping potatoes compared to red, yellow or speciality potatoes. Enhancing the concentrations and proportions of bioactive compounds with health benefits is an important consideration in maintaining and developing potato varieties for human consumption.

Kukoamines are organic compounds defined as catechol or caffeic acid derivatives of polyamines. They are found in potatoes, tomatoes and other solanaceous plants and are used in traditional Chinese herbal medicines [15, 25, 26]. Kukoamine A was measured in potatoes (*S. tuberosum* L.) [15, 24] and goji berries from *Lycium chinense* [27], and kukoamine B in gogi berries from *L. chinense* [26]. A recent review of the health benefits of these compounds [28] notes the association of jikoppi, the root bark of *L. chinense*, with hypertension, and refers to a short communication by Funayama et al. [22]. These researchers report that an intravenous administration of a crude methanol extract of *L. chinense* (0.5 g crude drug per kg) and a further refined extract comprising kukoamine A (5 mg per kg) induced hypotension in rats but no method description or data were presented. We have found no other references to support this bioactivity for kukoamine A or B. Evidence for the effect of potato consumption on BP is limited and contradictory. Vinson et al. [16] reported significant decreases in diastolic and systolic BP following a single meal of purple-skinned potatoes (~138 g) in healthy people while Borgi et al. [29] reported that higher intakes of baked, boiled or mashed potatoes and French fries were associated with an increased risk of developing hypertension.

We hypothesised that the oral administration of kukoamine A would reduce BP in a rat model of hypertension. The aim of this study was to firstly determine the safety of a single oral dose of synthetic kukoamine A (Fig 1) to mice and observed over 14 days using Organization for Economic Co-operation and Development (OECD) methodology. This methodology was developed to assess the acute toxicity of a test substance, and has been validated *in vivo*, uses very few animals and is reproducible. Secondly, we investigated whether daily oral doses of kukoamine A over a 5-week period reduced BP in a laboratory animal model of hypertension (Spontaneously Hypertensive Rats; SHR). The Wistar Kyoto (WKY) rats provide baseline measures for normal rat BP.

## Materials and methods

### Synthesis of kukoamine A

The synthesis of kukoamine A was carried out by a modification of the procedure in Piletska et al. [30]. A solution (2.18 g, 12 mmol) of 3,4-dihydroxyphenylpropionic acid (Sigma-Aldrich, USA) in 20 mL of tetrahydrofuran (THF; Fisher Chemicals, USA) was prepared at ambient

**Fig 1. Chemical structure of kukoamine A (C28H42N406** https://pubchem.ncbi.nlm.nih.gov/**).**

temperature. Dicyclohexylcarbodiimide (2.47 g, 12 mmol; Sigma-Aldrich, USA) and 4-dimethylaminopyridine (10 mg, 0.08 mmol; Sigma-Aldrich, USA) and then N-hydroxysuccinimide (1.38 g, 12 mmol; Sigma-Aldrich, USA) were then added sequentially. The reaction mixture was stirred for 18 h at ambient temperature then solids were removed by filtration and washed with THF (2 x 50 mL). This crude reaction product solution was treated with spermine (1.21 g, 6 mmol) in THF (5 mL). After the immediate formation of a white precipitate, the reaction was stirred for 4 h whereupon solids were filtered off and washed with THF (2 x 100 mL). Solvents were removed in vacuo and the crude material was purified on an SPE C18 prepacked column (Strata x 1g/20 mL Giga 8B-S100-JEG, Phenomenex, USA) conditioned with 20 mL of methanol (MeOH), then 0.2% formic acid in water (20 mL). A typical sample loading was 61 mg dissolved in 0.2% formic acid in water (2 mL). The column was eluted successively with 0.2% formic in water (2 x 20 mL), then with 20 mL volumes of 0.1% formic acid containing successively 10, 20, 30 and 40% MeOH, and finally with 100% MeOH (20 mL). The fractions containing the product, as analysed by LCMS and by comparison with a commercial standard (Kukoamine A, TransMIT, Giessen, Germany) were combined, the methanol removed in vacuo and then freeze-dried. The NMR analysis [22, 31] confirmed the product was kukoamine A with no kukoamine B detected. The model of NMR spectrometer used was a Bruker Avance 500. For $^1$H the operating frequency was 500.13 MHz. **$^1$H NMR** ($D_2O$) d 1.63 (8H, m), 2.50 (8H, m), 2.76 (4H, tr, $J$ = 7.5 Hz), 2.83 (4H, br, m), 3.13 (4H, tr, $J$ = 6.5 Hz), 6.64 (2H, dd, $J$ = 8, 2 Hz), 6.71 (2H, d, $J$ = 2 Hz), 6.79 (2H, d, $J$ = 8 Hz).

**Purification and gavage preparation.** The kukoamine preparation (~300 mg) was dissolved in 20 ml 1% (v/v) aqueous formic acid and sonicated. Six SPE columns (Strata X 1g/20 mL Giga 8B-S100-JEG), mounted over a common collection tank, were conditioned with 1% formic acid prior to loading with the kukoamine solution. Low-affinity compounds were eluted with 1% formic acid before the kukoamine was eluted with 10% (v/v) aqueous methanol. The columns were regenerated with 100% methanol before reconditioning with formic acid in preparation for the next run. Purity of the kukoamine fraction was confirmed by LCMS. The retention time of the synthesised kukoamine A was similar to the purchased standard and the accurate mass was measured at m/z 529.3067 [M-H]$^-$, which is consistent with the elemental formula of $C_{28}H_{42}N_4O_6$ (mDa +3.6, mSigma 13.0). Losses in the formic acid fraction and the methanol wash were minor. This method enabled us to produce in excess of 1 g kukoamine A.

The instability of the pure compound was an appreciable problem. Any moisture in contact with the freeze-dried powder resulted in rapid discolouration. This could be managed in solution by maintaining acidic conditions. We used formic acid for the purification step and citric acid (food grade, Hawkins Watts New Zealand (NZ), Auckland, NZ) for the gavage solution. We determined the molar extinction coefficient to be 938 at 260 nM, enabling spectrophotometric measurement to be used to check kukoamine concentration for preparation of the gavage solutions.

The purified kukoamine A was concentrated by rotary evaporation. The reduction was acidified by addition of citric acid and potassium citrate (food grade, Hawkins Watts NZ, Auckland, NZ) to provide an 8 mM citrate buffer pH 4.1. The concentration was checked by spectrophotometer and diluted with 8 mM citrate buffer as required to provide gavage solutions of suitable kukoamine A concentration for the treatments. These were frozen at -20˚C until required. The control gavage was 8 mM citrate buffer (pH 4.1).

## 14-day acute oral safety study

Eight-week-old female Swiss mice were used for the oral safety study. These tests were conducted according to the principles of the OECD method 407 limit test [32], whereby the number of animals is kept to the minimum required and the dose given is administered on a body weight basis. Animal procedures were approved by AgResearch Grasslands Animal Ethics Committee (Palmerston North) according to the Animal Welfare Act 1999, New Zealand (Applications 14237, 14363). The female, non-pregnant mice were supplied by Plant & Food Research, Food Evaluation Unit, Palmerston North, New Zealand. Lipnick et al. [33] recommended females be used in fixed-dose procedure studies as they are more sensitive. They were housed in family groups in a room maintained at 22 ± 3˚C, 30–70% humidity, 12 hr light:12 hr dark lighting, and fed a commercial pelleted diet (LabDiet) supplied by Fort Richards/Able Scientific. They were housed in their family groups from weaning at 3 weeks until 8 weeks of age and were weighed weekly.

Mice were given a single dose of either 5 or 10 mg kg$^{-1}$ bodyweight kukoamine A or citrate buffer (control) by oral gavage (less than 10 mL kg$^{-1}$ bodyweight) and observed for 14 days (n = 6 per treatment). We selected the mice for testing from those available of the appropriate age and weight range (±20% of the mean weight), marked them to allow for individual identification, and weighed and housed them in their dose groups (n = 3). Food was removed 1–2 h prior to the administration of the kukoamine dissolved in citrate buffer or the citrate buffer control, and was withheld for 1–2 h post-administration. Each mouse was observed for 30 min after dosing, thereafter periodically during the first 24 h with more frequent observation during the first 4 h, and daily thereafter for 14 days. If all three mice given either the control or kukoamine treatments showed no adverse effects for 48 h then a further three mice were given each treatment (n = 6). Mouse weights were recorded prior to substance administration, and weekly thereafter. At the end of the 14-day observation period, all animals were weighed and humanely killed. This acute procedure caused no ill effects in the mice given the two kukoamine A concentrations (5 or 10 mg kg$^{-1}$).

## Daily dose hypertension study

Male WKY and SHR rats, aged 6 weeks, were supplied by Animal Resources Centre, Perth, Australia. Animal procedures were approved by AgResearch Grasslands Animal Ethics Committee (Palmerston North), according to the Animal Welfare Act 1999, New Zealand (Application 14437). They were housed in a room maintained at 22 ± 3˚C, 30–70% humidity, 12 h light:12 h dark lighting. From six to eight weeks of age, the rats were housed in pairs and fed a commercial pelleted diet (LabDiet) supplied by Fort Richards/Able Scientific. The rats were then moved to individual cages for the duration of the study, and fed a powdered rodent diet (S1 Dataset). The rats were acclimated to the BP measuring equipment and procedures for two weeks prior to the start of the study. The rats (11 weeks of age) were randomly allocated to four experimental treatments (n = 10): control WKY, control SHR, SHR kukoamine A 5 mg kg$^{-1}$ and SHR kukoamine A 10 mg kg$^{-1}$. The control (citrate buffer) and kukoamine A

treatments were administered by daily oral gavage ($\leq$ 10 mL kg$^{-1}$ bodyweight) for five weeks (16 weeks of age). Bodyweights and BP measurements were recorded twice weekly.

## BP measurements

BP was measured using a non-invasive system tail-cuff method that uses volume-pressure recording (VPR) technology to detect changes in tail volume that correspond to systolic and diastolic BPs. The CODA non-invasive BP system was supplied by Kent Scientific (Torrington, CT, USA) with a set of cuffs and sensors (RAT-CUFFKIT, Kent Scientific) and can measure BP in up to eight animals simultaneously [34]. The VPR technology has been validated with simultaneous radiotelemetry BP measures [35]. Fifteen measurement cycles were performed for each animal at each time point. The data were cleaned by removing all readings with a blood volume less than 30 and data points that were greater than two standard deviations from the mean were excluded [35]. The remaining data were used to calculate the mean at each time point and the weekly value calculated from the two daily means taken during each week. All statistical analyses were carried out using Genstat (version 17, 2014, VSN Ltd, Hemel Hempstead, UK).

## Data analysis

Baseline food intakes and body weights were compared by analysis of variance. For the repeated measures data taken during the study, mixed-effects models were fitted with fixed effects for WKY vs SHR rats, diet (0, 5 or 10 mg kg$^{-1}$ kukoamine A) for the SHR animals, time, and interactions. The animal was fitted as a random effect. Several different correlations patterns between repeated observations on the same animal were tested, following Littell et al. [36]. For the food intake data, the simplest model (random effect for the animal, no autoregressive correlations, homogeneous residual variance) was chosen because it had the lowest Bayesian Information Criterion (BIC). For body weight, the best fitting model had unequal residual variance (lower in weeks 2–4 than 1, or 5 or final weight). For the BP data, the best fitting model had a random effect for the animal, no additional correlations between observations, and homogeneous residual variance.

## Results

No adverse effects were observed in the mice following the oral administration of kukoamine A at 5 and 10 mg kg$^{-1}$ bodyweight. The body weights and weight gains of the mice are shown in S1 Dataset. At the start of the study, the WKY rats had significantly lower body weights than the SHR rats (Table 1). All the rats gained weight during the study, and by the end of study, this difference was no longer significant. The food intakes of the WKY rats were lower than the SHR rats and remained constant over the five weeks of the study (Table 2). The food consumption of the SHR rats increased in weeks 3 and 4 but returned to the earlier amounts by the end of the study. The SHR rats had higher systolic BP at the beginning of the study as expected (Table 3). Systolic BP increased over the course of the study for both rat strains (Table 4). The BP of the SHR rats remained higher than the WKY rats but the pattern of change over time did not differ significantly between the experimental groups. Diastolic BP also increased over time for the SHR rats only (Table 5). The mean arterial BP (Table 6) increased over the course of the study for both rat strains. Among the SHR rats, there were no significant differences between groups fed different doses of kukoamine A on any of the measures.

**Table 1. Weekly mean body weights (g) of the rats on the experimental treatments[1].**

| Treatments | Week 1 | Week 2 | Week 3 | Week 4 | Week 5 | Final |
|---|---|---|---|---|---|---|
| WKY control | 222 ± 4 | 242 ± 4 | 258 ± 5 | 274 ± 5 | 285 ± 6 | 293 ± 6 |
| SHR control | 235 ± 2 | 255 ± 3 | 271 ± 3 | 282 ± 4 | 293 ± 4 | 300 ± 5 |
| SHR Kukoamine 5 | 239 ± 3 | 257 ±3 | 273 ± 4 | 287 ± 4 | 296 ± 4 | 305 ± 4 |
| SHR Kukoamine 10 | 235 ± 3 | 254 ± 3 | 270 ± 3 | 283 ± 3 | 290 ± 4 | 297 ± 3 |
| *Linear Mixed-effects model* | *P values* | | | | | |
| WKY vs SHR | 0.007 | | | | | |
| Diet within SHR | 0.772 | | | | | |
| Time | <0.001 | | | | | |
| Time x WKY x SHR | 0.005 | | | | | |
| Time x diet within SHR | 0.743 | | | | | |

[1] Body weights are expressed as mean ± standard error of the mean. Data were compared by analysis of variance. A least significant difference (lsd): Between Treatments lsd = 12; over time within treatment lsd = 4.

## Discussion

In the present study, we investigated the effect of ingesting synthetic kukoamine A on BP in a rat model of hypertension. Firstly, we determined that the ingestion of the synthetic kukoamine A at 5 and 10 mg kg$^{-1}$ was safe for mice. We then applied the SHR rat model of hypertension to determine if kukoamine A given orally would reduce BP at these doses. We found that the ingestion of kukoamine for five weeks at these doses had no effect on systolic BP in SHR rats.

We found no adverse effects on food intake or body weight in the mice and rats given the synthetic kukoamine A at 5 and 10 mg kg$^{-1}$ bodyweight by oral gavage. Similarly, no significant effects on body weight were observed for kukoamine B administered by daily gavage at the higher doses of 25 and 50 mg kg$^{-}$for 5 weeks and 9 weeks by Zhao et al. [37] and Li et al. [38] respectively. Kukoamine A and B are positional isomers differing only in the dihydrocaffeoylation position [39] so it has been assumed they will exhibit similar physiological responses. Li et al. [39], however, reported the cytoprotective effects of kukoamine B were superior to those of kukoamine A in Fenton-damaged bone marrow-derived mesenchymal stem cells.

**Table 2. Weekly mean food intakes (g) of the rats on the experimental treatments[1].**

| Treatments | Week 1 | Week 2 | Week 3 | Week 4 | Week 5 | Final |
|---|---|---|---|---|---|---|
| WKY control | 109 ± 2 | 108 ± 2 | 105 ± 2 | 108 ± 2 | 108 ± 3 | 105 ± 2 |
| SHR control | 125 ± 3 | 127 ± 2 | 133 ± 5 | 133 ± 3 | 130 ± 4 | 125 ± 3 |
| SHR Kukoamine 5 | 129 ± 2 | 128 ± 4 | 131 ± 4 | 134 ± 3 | 128 ± 3 | 129 ± 2 |
| SHR Kukoamine 10 | 125 ± 2 | 126 ± 2 | 129 ± 3 | 130 ± 3 | 120 ± 3 | 125 ± 2 |
| *Linear Mixed-effects model* | *P values* | | | | | |
| WKY vs SHR | <0.001 | | | | | |
| Diet within SHR | 0.460 | | | | | |
| Time | <0.001 | | | | | |
| Time x WKY x SHR | 0.034 | | | | | |
| Time x diet within SHR | 0.164 | | | | | |

[1] Food intakes are expressed as mean ± standard error of the mean. Data were compared by analysis of variance. A least significant difference (lsd): Between Treatments lsd = 8.8; over time within treatment lsd = 5.1.

Table 3. Systolic, diastolic and mean arterial blood pressure (BP) (mm Hg) of the rats at baseline[1].

| Treatments | Week 1 | Week 2 | Week 3 |
|---|---|---|---|
| WKY control | 156 ± 4[a] | 106 ± 3 | 122 ± 3 |
| SHR control | 187 ± 5[b] | 129 ± 4 | 148 ± 4 |
| SHR Kukoamine 5 | 184 ± 4 [b] | 130 ± 3 | 148 ± 3 |
| SHR Kukoamine 10 | 188 ± 2 [b] | 132 ± 2 | 150 ± 2 |
| Least significant difference | 12 | 9 | 9 |
| Time x diet within SHR | P<0.001 | P<0.001 | P<0.001 |

[1] BPs are expressed as mean ± standard error of the mean. Data were compared by analysis of variance.

The lack of response to the administration of kukoamine A on BP in the present study is in contrast to the earlier reports in which the intravenous administration of a crude methanol extract of *L. chinense* (0.5 g crude drug kg$^{-1}$) and a further refined extract comprising kukoamine A (5 mg kg$^{-1}$) induced hypotension in rats [22]. Kukoamine A administered at 5, 10 and 20 mg/kg intravenously has also been reported to reduce the neuro-inflammatory response and protect neurogenesis following whole-brain irradiation [40]. The doses of kukoamine A used here were based on those used previously, indicating that the method of administration (intravenous vs oral) may have influenced the bioavailability and therefore efficacy of these compounds. Administering the kukoamine A orally may have reduced or eliminated the kukoamine reaching the circulatory system due to the effects of the processes of digestion and absorption resulting in the complete or partial degradation of the kukoamine. We found no published information on the degradation and subsequent bioavailability of kukoamine following oral ingestion.

The SHR rat model of hypertension is characterised by a progressive increase in arterial BP with a rapid onset from 8 weeks of age, which is greater in males than females from 8 to 20 weeks of age [41]. This is caused by elevated peripheral vascular resistance, produced by neural and kidney factors causing structural vascular changes by elevated vascular protein synthesis. In the present study, BP was recorded via the non-invasive tail-cuff method, which has been validated against the gold standard invasive intravascular simultaneous radio telemetry [42]. The BPs measured in the present study were similar to those reported in other studies using this [43–46] and other hypertension models [47].

Table 4. Weekly mean systolic blood pressure (BP) (mm Hg) of the rats on the experimental treatments[1].

| Treatments | Week 1 | Week 2 | Week 3 | Week 4 | Week 5 | Final |
|---|---|---|---|---|---|---|
| WKY control | 158 ± 2 | 159 ± 4 | 158 ± 4 | 161 ± 4 | 165 ± 8 | 162 ± 4 |
| SHR control | 189 ± 2 | 192 ± 2 | 193 ± 2 | 203 ± 2 | 196 ± 4 | 202 ± 5 |
| SHR Kukoamine 5 | 192 ± 3 | 197 ± 3 | 196 ± 3 | 210 ± 2 | 211 ± 2 | 209 ± 4 |
| SHR Kukoamine 10 | 191 ± 3 | 193 ± 3 | 196 ± 3 | 203 ± 4 | 206 ± 3 | 200 ± 5 |
| *Linear Mixed-effects model* | | *P values* | | | | |
| WKY vs SHR | | <0.001 | | | | |
| Diet within SHR | | 0.201 | | | | |
| Time | | <0.001 | | | | |
| Time x WKY x SHR | | 0.252 | | | | |
| Time x diet within SHR | | 0.781 | | | | |

[1] BPs are expressed as mean ± standard error of the mean. Data were compared using mixed-effect models. A least significant difference (lsd): Between Treatments lsd = 11; over time within treatment lsd = 9.

**Table 5. Weekly mean diastolic blood pressure (BP) (mmHg) of the rats on the experimental treatments[1].**

| Treatments | Week 1 | Week 2 | Week 3 | Week 4 | Week 5 | Final |
|---|---|---|---|---|---|---|
| WKY control | 103 ± 3 | 105 ± 3 | 101 ± 2 | 104 ± 4 | 114 ± 6 | 105 ± 4 |
| SHR control | 133 ± 3 | 140 ± 3 | 144 ± 3 | 143 ± 5 | 146 ± 4 | 155 ± 4 |
| SHR Kukoamine 5 | 137 ± 5 | 143 ± 4 | 146 ± 4 | 154 ± 3 | 157 ± 4 | 156 ± 7 |
| SHR Kukoamine 10 | 139 ± 3 | 139 ± 4 | 145 ± 3 | 150 ± 5 | 155 ± 5 | 152 ± 5 |
| *Linear Mixed-effects model* | *P values* | | | | | |
| WKY vs SHR | <0.001 | | | | | |
| Diet within SHR | 0.417 | | | | | |
| Time | <0.001 | | | | | |
| Time x WKY x SHR | 0.083 | | | | | |
| Time x diet within SHR | 0.890 | | | | | |

[1] BPs are expressed as mean ± standard error of the mean. Data were compared using mixed-effect models. A least significant difference (lsd): Between Treatments lsd = 12; over time within treatment lsd = 11.

The primary cause of the development and occurrence of hypertension remains unclear but it is known that this disease is heterogeneous, polygenic, and multifactorial. BP is controlled by several complex physiological mechanisms including baroreflex and the renin-angiotensin system (RAS) [48, 49]. In a study investigating the effect of lentil varieties on vascular function in SHR and WKY rats [50], green and red lentils reduced arterial stiffness; however, BP was not affected. BP has been effectively lowered in this model of hypertension by the inhibition of the RAS, calcium antagonists, and by direct vasodilators [51]. Diuretics and endothelin antagonists are reported to be less effective and those of β-blockers were variable in this model. The potential mechanisms by which kukoamine influences BP have not yet been identified [51]. A limitation of this study is the absence of a positive control drug treatment in this study design. There have been a number of previous studies that have included drugs as a positive control to validate or apply this model [52–55]. We used the normotensive rat strain WKY from which the SHR rats were originally bred. The use of additional animals for another positive control treatment was not warranted. The present study determined only the physiological response (BP) of this animal model. Measuring biomarkers of vascular function such as gene and

**Table 6. Weekly mean arterial blood pressure (BP) (mm Hg) of the rats on the experimental treatments[1].**

| Treatments | Week 1 | Week 2 | Week 3 | Week 4 | Week 5 | Final |
|---|---|---|---|---|---|---|
| WKY control | 222 ± 4 | 242 ± 4 | 258 ± 5 | 274 ± 5 | 285 ± 6 | 293 ± 6 |
| SHR control | 235 ± 2 | 255 ± 3 | 271 ± 3 | 282 ± 4 | 293 ± 4 | 300 ± 5 |
| SHR Kukoamine 5 | 239 ± 3 | 257 ±3 | 273 ± 4 | 287 ± 4 | 296 ± 4 | 305 ± 4 |
| SHR Kukoamine 10 | 235 ± 3 | 254 ± 3 | 270 ± 3 | 283 ± 3 | 290 ± 4 | 297 ± 3 |
| *Linear Mixed-effects model* | *P values* | | | | | |
| WKY vs SHR | 0.007 | | | | | |
| Diet within SHR | 0.772 | | | | | |
| Time | <0.001 | | | | | |
| Time x WKY x SHR | 0.005 | | | | | |
| Time x diet within SHR | 0.743 | | | | | |

[1] Mean arterial BP = (diastolic BP + (systolic BP—diastolic BP)/3)); BPs are expressed as mean ± standard error of the mean. Data were compared using mixed-effect models. A least significant difference (lsd): Between Treatments lsd = 11; over time within treatment lsd = 10.

protein expression that play a role in mammalian blood pressure control or the hypertension pathway may have provided information on the potential mechanisms of kukoamine A.

The data presented here demonstrated there was no effect of orally administered synthetic kukoamine A on arterial hypertension. This was in contrast to the earlier study [22] using venous administration of kukoamine A extracted from *L. chinense* root barks. This may indicate that the bioavailability and bioactivity of kukoamine was affected by the processes of digestion and absorption. Oral treatments are preferred over injectable by patients for chronic conditions [56–58]. Investigations to confirm the results and clarify the methodology of the original study [22] reporting the hypotensive effect of kukoamine A are required. Subsequent research can then be undertaken to compare the responses of oral and intravenous administration of kukoamine A to understand the effects of gastrointestinal digestion and absorption on its impact on hypertension.

## Supporting information

**S1 Dataset. Master data.**
(XLSX)

## Acknowledgments

We thank Marian McKenzie and Nigel Perry for reviewing this manuscript.

## Author Contributions

**Conceptualization:** Christine A. Butts, Ross E. Lill.

**Data curation:** Christine A. Butts, Sheridan Martell, Hannah Dinnan.

**Formal analysis:** Christine A. Butts, Duncan I. Hedderley.

**Funding acquisition:** Ross E. Lill.

**Investigation:** Christine A. Butts, Sheridan Martell, Hannah Dinnan, Susanne Middlemiss-Kraak, Barry J. Bunn, Tony K. McGhie, Ross E. Lill.

**Methodology:** Christine A. Butts, Barry J. Bunn, Tony K. McGhie, Ross E. Lill.

**Project administration:** Christine A. Butts, Ross E. Lill.

**Resources:** Christine A. Butts, Tony K. McGhie, Ross E. Lill.

**Writing – original draft:** Christine A. Butts, Ross E. Lill.

**Writing – review & editing:** Christine A. Butts, Duncan I. Hedderley, Sheridan Martell, Hannah Dinnan, Susanne Middlemiss-Kraak, Barry J. Bunn, Tony K. McGhie, Ross E. Lill.

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
