## [Decision Letter · Decision Letter 0]

6 Dec 2021

PONE-D-21-35268 

Influence of oral administration of kukoamine A on blood pressure in a rat hypertension modelPLOS ONE

Dear Dr. Christine Ann Butts

Thank you for submitting your manuscript to PLOS ONE. After careful consideration, we feel that it has merit but does not fully meet PLOS ONE’s publication criteria as it currently stands. Therefore, we invite you to submit a revised version of the manuscript that addresses the points raised during the review process.

A marked-up copy of your manuscript that highlights changes made to the original version. You should upload this as a separate file labeled 'Revised Manuscript with Track Changes'.An unmarked version of your revised paper without tracked changes. You should upload this as a separate file labeled 'Manuscript'.

We look forward to receiving your revised manuscript.

Kind regards,

Balamuralikrishnan Balasubramanian

Academic Editor

PLOS ONE

Journal Requirements:

2. We note you have included a table to which you do not refer in the text of your manuscript. Please ensure that you refer to Table 5, and 6 in your text; if accepted, production will need this reference to link the reader to the Table.

Reviewers' comments:

Reviewer's Responses to Questions

**Comments to the Author**

1. Is the manuscript technically sound, and do the data support the conclusions?

Reviewer #1: Yes

Reviewer #2: Partly

2. Has the statistical analysis been performed appropriately and rigorously? 

Reviewer #1: Yes

Reviewer #2: Yes

3. Have the authors made all data underlying the findings in their manuscript fully available?

Reviewer #1: Yes

Reviewer #2: Yes

4. Is the manuscript presented in an intelligible fashion and written in standard English?

Reviewer #1: Yes

Reviewer #2: Yes

5. Review Comments to the Author

Reviewer #1: 1. The English need improvement since there are many grammatical and syntax errors in the manuscript. For example,

• in line number 16, the words “that have” may be as “have”;

• in line number 28, “a rat” as “rat”;

• in line number 79, “given” as “were given”;

• in line number 81, “acute” as “the acute”;

• in line number 97, “immediate” as “the immediate”;

• in line number 104, “product” as “the product”;

• in line number 191, 193 and 197, “Animal” as “The animal”;

• in line number 254, “those” as “to those”;

• in line number 256, “administration” as “the administration”;

• in line number 275, “study were” as “study was”;

• in line number 287, “pressure have” as “pressure has”;

• in line number 290, “positive” as “a positive”.

The grammar mistakes which are not mentioned here are also to be checked and corrected properly.

2. There are some typing mistakes as well, and authors are advised to carefully proof-read the text. For example,

• in line number 34, the words “pressure lowering” may be as “pressure-lowering”;

• in line number 35 and 36, “angiotension” as “angiotensin”;

• in line number 36, “beta blockers” as “beta-blockers”;

• in line number 45, “nitrate containing” as “nitrate-containing”;

• in line number 74, “purple skinned” as “purple-skinned”;

• in line number 85, “strain are” as “strains are”;

• in line number 115, “Low affinity” as “Low-affinity”;

• in line number 151, “post administration” as “post-administration”;

• in line number 174, “tail cuff” as “tail-cuff”;

• in line number all over the manuscript, “mixed effects” as “mixed-effects”;

• in line number 191, “correlation” as “correlations”;

• in line number 203, “of study” as “of study,”;

• in line number 236, “pressue” as “pressure”;

• in line number 261, “whole brain” as “whole-brain”.

The typos not mentioned here are also to be checked and corrected properly.

3. Check the abbreviations throughout the manuscript and introduce the abbreviation when the full word appears the first time in the text and then use only the abbreviation (For example, blood pressure (BP), renin-angiotensin system (RAS), etc.,). And it should be in both abstract as well as in the remaining part of the manuscript.

4. The full form of the species should be given when the first time appears and followed by only the first letter of the genus in both the abstract and the remaining part of the manuscript (e.g., Solanum tuberosum; Lycium chinense when the first time appears and followed by S. tuberosum; L. chinense).

5. In the materials and methods, the author should include the source of chemicals used in this study. And also the age of the animal model chosen may be given in the materials and methods section.

6. In the result tables, when writing the significance, authors are not mentioned what is a,b,c, which are given in the superscript of values. It may be rewrite as values not sharing a common marking (a, b, c,..) differ significantly at P<0.05 or other format.

7. The table legends should be improved and a proper footnote should be given. All legends should have enough description for a reader to understand the figure without having to refer back o the main text of the manuscript.

8. The authors have mentioned that the fractions containing product was analysed by LCMS and by comparison with a commercial standard. But in the results section the results is not given and also in materials the commercial standard tried is not given. It should be given properly (in line number 105) and also in results section.

9. Similarly, the authors have given NMR analysis are also done but it is not given in the results section and it should be given properly (in line number 106) and also in results section.

10. The author should justify how the dose fixation an duration study are done (is it based on toxicity test or previous reference, if previous should be included in the reference section) in the discussion part of the manuscript.

Reviewer #2: This topic “Influence of oral administration of kukoamine A on blood pressure in a rat hypertension model” is an interesting work and quite good information.

However, before accepting publication, minor revision is required and some points in this manuscript need to clarify. So, the comment and suggestion to the author is a PDF attachment file, please check it out.

6. PLOS authors have the option to publish the peer review history of their article (what does this mean?). If published, this will include your full peer review and any attached files.

Reviewer #1: **Yes: **Prof. A> Vijaya Anand

Reviewer #2: **Yes: **UTTHAPON ISSARA

---

## [Author Response · Author response to Decision Letter 0]

1 Mar 2022

Firstly, our thanks to the reviewers for their suggestions, time, knowledge and interest in this manuscript.

The journal template has been used and the formatting checked during the revision.

2. We note you have included a table to which you do not refer in the text of your manuscript. Please ensure that you refer to Table 5, and 6 in your text; if accepted, production will need this reference to link the reader to the Table.

Thank you for noting this error in the text. This has now been revised and the tables are referred to in the Results section (Lines 217-218, revised manuscript with track changes accepted).

We have checked the references and made the appropriate revisions prior to resubmission.

Reviewer #1: 

1. The English need improvement since there are many grammatical and syntax errors in the manuscript.

• in line number 16, the words “that have” may be as “have”;

• in line number 28, “a rat” as “rat”;

• in line number 79, “given” as “were given”;

• in line number 81, “acute” as “the acute”;

• in line number 97, “immediate” as “the immediate”;

• in line number 104, “product” as “the product”;

• in line number 191, 193 and 197, “Animal” as “The animal”;

• in line number 254, “those” as “to those”;

• in line number 256, “administration” as “the administration”;

• in line number 275, “study were” as “study was”;

• in line number 287, “pressure have” as “pressure has”;

• in line number 290, “positive” as “a positive”.

The grammar mistakes which are not mentioned here are also to be checked and corrected properly.

Authors’ response

The grammatical and syntax errors have been amended throughout the manuscript and independently reviewed as well. Thank you.

2. There are some typing mistakes as well, and authors are advised to carefully proof-read the text. For example,

• in line number 34, the words “pressure lowering” may be as “pressure-lowering”;

• in line number 35 and 36, “angiotension” as “angiotensin”;

• in line number 36, “beta blockers” as “beta-blockers”;

• in line number 45, “nitrate containing” as “nitrate-containing”;

• in line number 74, “purple skinned” as “purple-skinned”;

• in line number 85, “strain are” as “strains are”;

• in line number 115, “Low affinity” as “Low-affinity”;

• in line number 151, “post administration” as “post-administration”;

• in line number 174, “tail cuff” as “tail-cuff”;

• in line number all over the manuscript, “mixed effects” as “mixed-effects”;

• in line number 191, “correlation” as “correlations”;

• in line number 203, “of study” as “of study,”;

• in line number 236, “pressue” as “pressure”;

• in line number 261, “whole brain” as “whole-brain”.

The typos not mentioned here are also to be checked and corrected properly.

Authors’ response

The typing mistakes have been amended throughout the manuscript. Thank you.

3. Check the abbreviations throughout the manuscript and introduce the abbreviation when the full word appears the first time in the text and then use only the abbreviation (For example, blood pressure (BP), renin-angiotensin system (RAS), etc.,). And it should be in both abstract as well as in the remaining part of the manuscript.

Authors’ response

The abbreviations have been amended throughout the manuscript as requested.

4. The full form of the species should be given when the first time appears and followed by only the first letter of the genus in both the abstract and the remaining part of the manuscript (e.g., Solanum tuberosum; Lycium chinense when the first time appears and followed by S. tuberosum; L. chinense).

Authors’ response

The abbreviations have been amended throughout the manuscript as requested.

5. In the materials and methods, the author should include the source of chemicals used in this study. And also the age of the animal model chosen may be given in the materials and methods section.

Authors’ response

The sources of the chemicals have been included as requested. The age of the rat model during the study has been added (Lines 175 and 179 in revised manuscript track changes accepted).

6. In the result tables, when writing the significance, authors are not mentioned what is a,b,c, which are given in the superscript of values. It may be rewrite as values not sharing a common marking (a, b, c,..) differ significantly at P<0.05 or other format.

Authors’ response

Thank you for the suggestion. Assisting the reader to understand the statistically significant effects is important. However, adding superscripts would clutter the table and create confusion as one set of superscripts are to be read down the table vertically (WHR vs SHR effect) while other superscripts would need to be applied and read horizontally. The biometrician (DIH) has recommended the current approach.

7. The table legends should be improved and a proper footnote should be given. All legends should have enough description for a reader to understand the figure without having to refer back o the main text of the manuscript.

Authors’ response

Further detail has been added as requested to all the table legends and footnotes.

8. The authors have mentioned that the fractions containing product was analysed by LCMS and by comparison with a commercial standard. But in the results section the results is not given and also in materials the commercial standard tried is not given. It should be given properly (in line number 105) and also in results section.

Authors’ response

Thank you for this suggestion. We have now included the details of the commercial standard and the results of the LCMS analysis (Lines 121-125, revised manuscript with track changes accepted).

9. Similarly, the authors have given NMR analysis are also done but it is not given in the results section and it should be given properly (in line number 106) and also in results section.

Authors’ response

We have now included the details of the commercial standard and the results of the NMR analysis (Lines 102-108, revised manuscript with track changes accepted).

10. The author should justify how the dose fixation an duration study are done (is it based on toxicity test or previous reference, if previous should be included in the reference section) in the discussion part of the manuscript.

Authors’ response

We undertook the oral safety of the kukoamine A using the material we had prepared to ensure there were no unexpected contaminants that could have negative effects on the animals. We applied the methodology endorsed by the OECD in their acute oral toxicity test (OECD 423) in which the compounds are given and the animals observed for 14 days. Other studies have successfully given kukoamine orally and intravenously so we did not undertake the more extensive testing as its intrinsic safety had been established in those studies [1-5]. Therefore, undertaking the more extensive longer-term study and blood sampling in more animals was not warranted. 

Reviewer #2: This topic “Influence of oral administration of kukoamine A on blood pressure in a rat hypertension model” is an interesting work and quite good information.

However, before accepting publication, minor revision is required and some points in this manuscript need to clarify. So, the comment and suggestion to the author is a PDF attachment file, please check it out.

1. “For 14-day acute oral safety study section: Why the researcher uses female mice to study this effect? Generally, the animal’s sexuality has influenced the genetic transformation or variance of a hormone than a male animal, so it will be difficult to control that factor and may influence to blood pressure determination? 

However, researcher use the male or female mice in this study? Because in Line 145 mentioned difference mice gender?”

Authors’ response

The purpose of the first part of this study was to determine the oral safety of the chemically synthesised kukoamine A using OECD testing guidelines and procedures. We used female mice based on the review by Lipnick et al. [6] who recommended the use of females only as they were more sensitive in acute toxicity studies. By using only one gender we were able to reduce the number of animals being manipulated. We also ensured they were not pregnant by separating the males and females at weaning as this could influence their physiological and metabolic responses. For this part of the study we only used adult female mice (non-pregnant).

Reviewer

2. “Also, the time of oral test is quite short time?? Only 2 weeks of an animal experiment may not be enough to get the cover results about the blood parameter index changing?

Authors’ response

We undertook the oral safety of the kukoamine A using the material we had prepared to ensure there were no unexpected contaminants that could have negative effects on the animals. We applied the methodology endorsed by the OECD in their acute oral toxicity test (OECD 423) in which the compounds are given and the animals observed for 14 days. Other studies have successfully given kukoamine orally and intravenously so we did not undertake the more extensive testing as its safety had been established in those studies [1-5]. Therefore, undertaking the more extensive longer-term study and blood sampling in more animals was not warranted. 

Reviewer

3. “In discussion point Line 250, previous studies already showed that higher dose oral test in mice 25 and 50 mg-1 was safe, so why researcher try to study the only a lower concentration of kukoamine A?? in which 5 and 10 mg kg-1 seems to be safe for sure”

Authors’ response

We agree that the earlier studies using doses of 25 and 50 mg of kukoamine B did show they were safe. However, we wanted to ensure the kukoamine A we had synthesized was safe when administered orally compared to intravenously and at the same concentration (5 mg per kg bodyweight) [3]. And secondly to investigate concentrations of kukoamines that are likely to be provided in whole foods such as potatoes [7]. 

Reviewer

4. “Check all the error typing and make a correction”

Authors’ response

Thank you, we have thoroughly checked the manuscript and amended the manuscript.

Reviewer

5. “Suggestion: The present study was determined only physiological and blood characteristics of the animal model, however, to confirm the mechanisms of effects of kukoamine A on vascular function, other biomarkers need to determine such as the expression of the gene-related vascular system and the significant protein expression that play a role to control the blood pressure or hypertension pathway in the mammalian.”

Authors’ response

Thank you for your suggestion. We have incorporated this in the Discussion lines 306-309 of the amended manuscript.

6. PLOS authors have the option to publish the peer review history of their article (what does this mean?). If published, this will include your full peer review and any attached files.

Authors’ response

Yes

References

1. Chaparro, J.M.; Holm, D.G.; Broeckling, C.D.; Prenni, J.E.; Heuberger, A.L. Metabolomics and Ionomics of Potato Tuber Reveals an Influence of Cultivar and Market Class on Human Nutrients and Bioactive Compounds. Frontiers in Nutrition 2018, 5, 36-36, doi:10.3389/fnut.2018.00036.

2. Funayama, S.; Yoshida, K.; Konno, C.; Hikino, H. Validity of oriental medicines .21. structure of kukoamine-a, a hypotensive principle of lycium-chinense root barks. Tetrahedron Lett. 1980, 21, 1355-1356, doi:10.1016/s0040-4039(00)74574-6.

3. Funayama, S.; Yoshida, K.; Konno, C.; Hikino, H. Structure of kukoamine A, a hypotensive principle of Lycium chinense root barks. Tetrahedron Lett. 1980, 21, 1355-1356.

4. Funayama, S.; Zhang, G.-R.; Nozoe, S. Kukoamine B, a spermine alkaloid from Lycium chinense. Phytochemistry 1995, 38, 1529-1531, doi:10.1016/0031-9422(94)00826-F.

5. Wang, L.; Wang, P.; Wang, D.; Tao, M.; Xu, W.; Olatunji, O.J. Anti-Inflammatory Activities of Kukoamine A From the Root Bark of Lycium chinense Miller. Natural Product Communications 2020, 15, 1934578X20912088, doi:10.1177/1934578x20912088.

6. Lipnick, R.L.; Cotruvo, J.A.; Hill, R.N.; Bruce, R.D.; Stitzel, K.A.; Walker, A.P.; Chu, I.; Goddard, M.; Segal, L.; Springer, J.A., et al. Comparison of the up-and-down, conventional LD50, and fixed-dose acute toxicity procedures. Food Chem. Toxicol. 1995, 33, 223-231, doi:10.1016/0278-6915(94)00136-C.

7. Vinson, J.A.; Demkosky, C.A.; Navarre, D.A.; Smyda, M.A. High-Antioxidant Potatoes: Acute in Vivo Antioxidant Source and Hypotensive Agent in Humans after Supplementation to Hypertensive Subjects. J. Agric. Food Chem. 2012, 60, 6749-6754, doi:10.1021/jf2045262.

---

## [Decision Letter · Decision Letter 1]

12 Apr 2022

Influence of oral administration of kukoamine A on blood pressure in a rat hypertension model

PONE-D-21-35268R1

Dear Dr. Christine Ann Butts

We’re pleased to inform you that your manuscript has been judged scientifically suitable for publication and will be formally accepted for publication once it meets all outstanding technical requirements.

Kind regards,

Balamuralikrishnan Balasubramanian

Academic Editor

PLOS ONE

Additional Editor Comments (optional):

Reviewers' comments:

Reviewer's Responses to Questions

**Comments to the Author**

1. If the authors have adequately addressed your comments raised in a previous round of review and you feel that this manuscript is now acceptable for publication, you may indicate that here to bypass the “Comments to the Author” section, enter your conflict of interest statement in the “Confidential to Editor” section, and submit your "Accept" recommendation.

Reviewer #1: All comments have been addressed

Reviewer #2: All comments have been addressed

2. Is the manuscript technically sound, and do the data support the conclusions?

Reviewer #1: Yes

Reviewer #2: Yes

3. Has the statistical analysis been performed appropriately and rigorously? 

Reviewer #1: Yes

Reviewer #2: Yes

4. Have the authors made all data underlying the findings in their manuscript fully available?

Reviewer #1: Yes

Reviewer #2: Yes

5. Is the manuscript presented in an intelligible fashion and written in standard English?

Reviewer #1: Yes

Reviewer #2: Yes

6. Review Comments to the Author

Reviewer #1: 1. There are some grammatical, alignment and typographical errors are noted in the manuscript and it should be thoroughly checked and corrected throughout the manuscript. For example,

• in line number 42, the word “include” may be as “including”;

• in line number 54, “specialty” as “speciality”;

• in line number 105, “NMR” as “the NMR”;

• in line number 147, “fixed dose” as “fixed-dose”;

• in line number 198, “correlation” as “correlations”;

• in line number 211, “study” as “study,”;

• in line number 224, 229, 239 and 245, “Least” as “A least”;

• in line number 251, “mixed effect” as “mixed-effect”;

• in line number 304, “instead” as “instead of”.

Reviewer #2: All points of wondering have been answered and clarified. So, this manuscript could be published and it can be useful for further study.

7. PLOS authors have the option to publish the peer review history of their article (what does this mean?). If published, this will include your full peer review and any attached files.

Reviewer #1: **Yes: **Prof. A. Vijaya Anand

Reviewer #2: **Yes: **UTTHAPON ISSARA

---

## [Editor Report · Acceptance letter]

21 Apr 2022

PONE-D-21-35268R1 

Influence of oral administration of kukoamine A on blood pressure in a rat hypertension model 

Dear Dr. Butts:

I'm pleased to inform you that your manuscript has been deemed suitable for publication in PLOS ONE. Congratulations! Your manuscript is now with our production department. 

Kind regards, 

on behalf of

Dr. Balamuralikrishnan Balasubramanian 

Academic Editor

PLOS ONE